# Supercritical CO_2_ Fluid Extraction of *Elaeagnus mollis* Diels Seed Oil and Its Antioxidant Ability

**DOI:** 10.3390/molecules24050911

**Published:** 2019-03-05

**Authors:** Chengxin Wang, Zhenhua Duan, Liuping Fan, Jinwei Li

**Affiliations:** 1Institute of Food Research, Hezhou University, Hezhou 542899, China; wcx18861502027@163.com; 2School of Food Science and Technology, Jiangnan University, 1800 Lihu Avenue, Wuxi 214122, China; jwli@jiangnan.edu.cn

**Keywords:** *Elaeagnus mollis* Diels seed, supercritical carbon dioxide extraction, orthogonal optimization, fatty acid composition, antioxidant activity

## Abstract

Supercritical fluid carbon dioxide (SF-CO_2_) was used to extract oil from *Elaeagnus mollis* Diels (*E. mollis* Diels) seed and its antioxidant ability was also investigated. The effect of extraction pressure (20–35 MPa), extraction temperature (35–65 °C), extraction time (90–180 min) and seed particle size (40–100 mesh) on the oil yield were studied. An orthogonal experiment was conducted to determine the best operating conditions for the maximum extraction oil yield. Based on the optimum conditions, the maximum yield reached 29.35% at 30 MPa, 50 °C, 150 min, 80 mesh seed particle size and 40 g/min SF-CO_2_ flow rate. The *E. mollis* Diels seed (EDS) oil obtained under optimal SF-CO_2_ extraction conditions had higher unsaturated fatty acid content (91.89%), higher vitamin E content (96.24 ± 3.01 mg/100 g) and higher total phytosterols content (364.34 ± 4.86 mg/100 g) than that extracted by Soxhlet extraction (SE) and cold pressing (CP) methods. The antioxidant activity of the EDS oil was measured by DPPH and hydroxyl radical scavenging test. EDS oil extracted by different methods exhibited a dose-dependent antioxidant ability, with IC_50_ values of no significant differences. Based on the results of correlation between bioactive compounds, lupeol and γ-tocopherol was the most important antioxidant in EDS oil.

## 1. Introduction

Oils are an essential part of the human daily diet and play an important role in human health. There are a variety of beneficial components in vegetable oils such as unsaturated fatty acids, fat-soluble vitamins, squalene and phytosterols, which cannot be produced by the human body. In order to meet the human nutrition demands, many plant seeds have been investigated as sources of vitamins and essential fatty acids [1].

*E. mollis* Diels belongs to the *Elaeagnaceae* family, and the plant is endemic to China [2]. *E. mollis* Diels is also regarded as an important economic plant because its seeds contain a large percentage of oil and protein [3]. Previous studies have indicated that *E. mollis* Diels seed (EDS) is a good edible oil resource with high vitamin E levels and antioxidant activities [4]. Thus, due to its composition and related properties, EDS oil could be used in pharmaceutical and functional food applications. Recently, many attempts have been made to investigate the health benefits of EDS oil such as hypolipidemic, antioxidant, anti-inflammatory and neuroprotective activity. Therefore, EDS has drawn increasing interest in recent years as a promising oil resource. However, no reports were found in the literature on the oil content and the composition of EDS extracted by supercritical CO_2_ fluid (SF-CO_2_). 

Oil seeds of plants are usually extracted with organic solvents such as *n*-hexane, ethyl ether and petroleum ether or mechanical pressing. However, solvent extraction requires long time, and the following separation at high temperature has negative impact on oil quality, especially the sensory quality. Mechanical pressing methods have limited application because the oil yield is low, although oil quality is great [5,6]. Supercritical fluid extraction (SFE) is known as a fast and efficient method for the extraction of various compounds from plant materials. SF-CO_2_ is the most common supercritical solvent since it is odorless, colorless, safe, inexpensive, nontoxic, recyclable and environment-friendly. In addition, CO_2_ can evaporate immediately when exposed to atmospheric conditions, thus the extract of SF-CO_2_ is free from chemicals and thermal degradation compounds [7]. The SFE technology has been applied in the extraction of plant oil from many plant materials such as grape seed [8], *Ganoderma lucidum* spore [9], sea buckthorn [10] and sunflower seed [11] in medical and functional food processing with great potential for further applications [12,13]. 

Optimization of the extraction conditions is a critical step to develop a successful process. Selection of diverse variables including extraction pressure, temperature, time and pretreatment of raw materials is an important factor affecting the final composition of the extract and the process efficiency of SFE. In addition, the dissolving capacity of SF-CO_2_ is changing with the variation of the extraction pressure and/or temperature, which means that SFE is a tailored extraction method for desired compounds while leaving undesirable compounds behinds [14]. In order to obtain optimal SFE conditions for EDS oil extraction, we have employed the orthogonal array design (OAD) procedure. OAD is one of the simple and systematic approaches for designing an experiment with less experimental runs [15]. Thus, OAD is possible to reduce the time and cost for the experimental investigations [16,17,18].

The objective of this study is to investigate the effect of extraction pressure, temperature, time and seed particle size on the EDS oil yield. OAD was employed to optimize extraction conditions for the maximum EDS oil yield. Furthermore, the fatty acid composition and antioxidant activity of the EDS oil obtained under optimized conditions were determined and compared with those obtained by the SE and CP method. The antioxidant activity of extracted oil was determined by the 2,2-diphenyl-1-picrylhydrazyl (DPPH) radical scavenging assay and hydroxyl (•OH) radical scavenging test. The overall aim of the study was to provide a reliable and theoretical basis for the commercial exploitation of EDS.

## 2. Results

### 2.1. Effects of SFE Conditions on Oil Yield

The effects of four SF-CO_2_ extraction parameters, namely extraction pressure (A), temperature (B), time (C) and particle size (D) on the oil yield were investigated by single factor experiment method. As shown in Figure 1, with the increase of extraction pressure from 20 to 35 MPa, the oil yield improved significantly (*p* < 0.05). The pressure above 35 MPa would lead to more energy consumption and less equipment safety. Thus, the extraction pressure was set from 30 to 35 MPa. The oil yield was the highest when the extraction time and temperature was 150 min, 55 °C respectively. Considering the SFE efficiency, energy consumption, and equipment security, 150 min and 55 °C were deemed the best extraction time and temperature for the optimization of SFE conditions. The oil yield increased with the decrease of particle size. However, if the particle size was too small, the equipment pipeline would be jammed seriously result in the decrease of EDS oil yield. Thus, the 80 mesh was deemed best for the optimization of SFE conditions. 

### 2.2. Optimization of the SFE Conditions

Based on the OA_9_ matrix nine experiments were conducted, and the corresponding oil yields are shown in Table 1. The assignment of the factors and their levels is also shown in Table 1. For each factor, the mean values of K at different levels and the range (R) were also shown in Table 1. It was used to evaluate the effects of each factor on oil yield. The highest extraction yield was obtained in run number 6 (28.02%) at 32.5 MPa, 50 °C, 180 min and 100 mesh. Moreover, in run number 9, the extraction yield was the least (24.29%) at 35 MPa, 60 °C, 120 min and 100 mesh. The oil yield varied significantly with the changing of operating conditions. Statistical analysis showed that the effects of various factors on the extraction process followed the order C > D > A > B; the effect of extraction time on the extraction process was the greatest, followed by particle size, extraction pressure and then extraction temperature. 

The variance analysis in Table 2 revealed that the extraction time significantly affected the EDS oil yield. The best extraction conditions were approximated by taking the level of each factor giving the maximum extraction. The optimum SFE parameters for achieving the maximum extraction yield of EDS oil were 30 MPa, 50 °C, 150 min and 80 mesh. Under such optimal SFE conditions, the extraction yield of EDS oil was tested by confirmation experiments repeated three times. The practical EDS oil yield was 28.32 ± 1.155%, which indicated that the process was reliable and effective. 

### 2.3. Identification and Composition of Fatty Acid in EDS Oils

The fatty acid analysis of EDS oil performed by GC-FID showed that EDS oil has a high content of linoleic acid, oleic acid, linolenic acid, as shown in Table 3. Kan et al. reported that the main fatty acids present in EDS oil were linoleic acid (44.7–54.8%), oleic acid (33.6–40.6%) and palmitic acid (3.1–5.5%) [3]. However, some minor differences in fatty acids were observed, that may depend on many factors, such as the geographical origin, climate, soil composition and especially extraction method. The unsaturated fatty acids (UFAs) extracted by optimized SC-CO_2_ extraction (91.89% of total fatty acid) was higher than those extracted by SE (85.16% of total fatty acid) and CP (90.92% of total fatty acid). Although the composition of fatty acids varied among the three extraction methods, the content of UFAs was approximately 90% which means the EDS oil must be prevented from contacting with air to oxidation when it is stored. This work showed that EDS oil is a great source of linoleic acid (C18:2 n-9, 12) and oleic acid (C18:1 n-9), accounting for up to 50% and 30% of total oil which obtained by different extraction process. The content of essential fatty acids (EFAs), namely linoleic acid (C18:2 n-9, 12) and linolenic acid (C18:3 n-9, 12, 15) in EDS oil obtained by optimized SF-CO_2_ extraction method was the highest. These two EFAs are necessary to support normal cell functions, promote disease prevention and may even be useful for disease treatment [19]. The compositions of fatty acids were different with the types of edible oils. However, the composition of fatty acids in EDS oil was similar with that in walnut oils from Serbia. It was reported that linoleic acid (57.2–65.1%), oleic acid (15.9–23.7%), linolenic acid (9.1–13.6%) and palmitic acid (6.3–7.7%), and the PUFAs, MUFAs and SFAs in walnut oil ranged in 67.4–75.7%, 15.9–23.9% and 8.3–9.4%, respectively [20]. Due to the better nutritional and health effects of EDS oil, it has great research and development value.

### 2.4. Tocopherols and Tocotrienols Content

Vitamin E is considered as the major lipid soluble antioxidant in vegetable oil, which is particularly important for prevention of diseases. It contains two groups of molecules: four tocopherols (α, β, γ and δ) and four tocotrienol isoforms (α, β, γ and δ) [21].

Table 4 reported the concentration of tocopherols and tocotrienols in EDS oil samples obtained by three methods. There were three isoforms (α, γ and δ) of tocopherols and tocotrienols in EDS oils. In all oil samples, the content of tocopherols, which accounted for 85% of total content, was higher than the content of tocotrienols. The total tocopherols and tocotrienols content of SF-CO_2_-extracted EDS oil was 96.24 ± 3.01 mg/100 g, which was the highest among the three extraction methods. γ-Tocopherol was the richest tocopherol, while δ-tocopherol was the lowest. The content of α-tocopherol in EDS oils was almost twice that of δ-tocopherol. These results were different with data previously reported by Liang et al., who reported that the total content of tocopherols ranged from 119.6 to 128.60 mg/100 g [4]. The difference may be caused by cultivars and different detection approach.

Besides, it was the first time to determine the content of tocotrienols in EDS oils. The concentration of total tocotrienols in EDS oils extracted by different methods had no significant differences. However, there were some differences among the content of each tocotrienol in the EDS oils obtained by different extraction methods. SF-CO_2_ extracted EDS oil was enriched in γ-tocotrienol, followed by α-tocotrienol and δ-tocotrienol. For EDS oils extracted by SE or CP, α-tocotrienol content was the highest followed by δ-tocotrienol and γ-tocotrienol.

These results represented that EDS oil is a great source of tocopherols and tocotrienols which compared with olive oil (17 mg/100 g) and soybean oil (49.7 mg/100 g) [22]. Each content of tocopherols and tocotrienols in EDS oil had some differences related to different extraction method. The results also indicated that SF-CO_2_ extraction was superior to Soxhlet-extracted and Cold-pressed in extraction of tocopherols and tocotrienols. 

### 2.5. Phytosterols

As shown in Table 5, four types of phytosterols were determined in EDS oils including β-sitosterol, stigmasterol, β-amyrin and lupeol. β-Sitosterol is the major phytosterol found in EDS oils, constituting 56–61% of total phytosterols among oils and its content in SF-CO_2_ extracted EDS oil was about two times higher than that in cold-pressed EDS oil. Lupeol is the second richest phytosterol accounting for 15–22% of total phytosterols in EDS oils which was found for the first time in EDS oils. Lupeol is one of the triterpenoids which exhibit a broad spectrum of relevant clinical activities, such as antihypertensive, antidiabetic, anti-inflammatory, anticancer and antimicrobial [23]. Compared with the literature that its content in EDS oil was about 5–6 times higher than that in sea buckthorn seeds oil [24].

Moreover, the total phytosterol contents were the highest in the SF-CO_2_ extracted EDS oil, followed by Soxhlet-extracted and Cold-pressed EDS oil, which were all higher than that in olive oil (144 mg/100 g) [25], coconut oil (114 mg/100 g) [26] and walnut oil (160 mg/100 g) [27]. The same trend was also found in the extraction of grape seed oils [28]. Many research results have shown that phytosterols can decrease the serum cholesterol levels in human body [29]. Therefore, the high levels of phytosterols in EDS oils implied that it could be used as healthy edible oils or developed as functional food ingredients.

### 2.6. Comparison of Antioxidant Activities of EDS Oil Extracted by Different Method

The free radical scavenging capacity of EDS oil extracted by different methods was assessed by DPPH and hydroxyl radical assays. As shown in Figure 2, the antioxidant activity of EDS oils was dose-dependent with the experimental concentration range both in DPPH and hydroxyl radical assays. This indicated that DPPH and hydroxyl radical scavenging activity was proportional to the increase of sample concentration. For DPPH radical scavenging activity, the IC_50_ (half maximal effective concentration) of the SF-CO_2_ extracted EDS oil, Soxhlet extracted EDS oil and cold-pressed EDS oil were 12.19 ± 2.01, 8.87 ± 0.96 and 10.18 ± 1.12 mg/mL, respectively. The DPPH radical scavenging capacity was as the following of Soxhlet extracted > cold-pressed > SF-CO_2_ extracted EDS oil. However, the IC_50_ of the SF-CO_2_ extracted EDS oil, Soxhlet-extracted EDS oil and cold-pressed EDS oil were 2.10 ± 0.27, 1.94 ± 0.57 and 1.42 ± 0.11 mg/mL for hydroxyl radical scavenging activity. The hydroxyl radical scavenging capacity was as the following of cold-pressed > Soxhlet extracted > SF-CO_2_ extracted EDS oil. The IC_50_ of EDS oils extracted by different methods on DPPH radical was higher than that of on hydroxyl radical. In addition, these results shown that the antioxidant capacities between EDS oils obtained by different methods had no significant difference.

The antioxidant properties of EDS oils have been described by some authors. Kan et al. [3] investigated the radical scavenging activities of three kinds of EDS oils by DPPH assay, the cold-pressed EDS oil showed the highest radical scavenging effects (around 60% when the concentration was 20 mg/mL). These results are different our study, the Soxhlet-extracted EDS oil showed the antioxidant activities valued 75.38% in DPPH assay under the same concentration. These differences would be explained by the different regions of raw materials and the different chemical compositions of EDS oil obtained by different extraction methods.

Taking in account the results of both DPPH and hydroxyl radical assays, it is possible to conclude that the EDS oils extracted by different methods both exhibited good antioxidant activities, especially under higher concentration. These results demonstrated that EDS oil could be widely used as a promising functional edible oil.

To correlate the antioxidant results between DPPH and hydroxyl radical assays, a regression analysis was significant correlation between DPPH and hydroxyl radical assays (R^2^ = 0.88415), which indicated that both DPPH and hydroxyl radical assays could be used to investigate the antioxidant activity of EDS oils [30,31].

According to many references, the antioxidant activity of some vegetable oils might be related to their natural antioxidant concentration, especially lipophilic antioxidant compounds such as tocopherols, tocotrienols and phytosterols with antioxidant activity (in both the DPPH and hydroxyl radical assays). A Pearson correlation analysis was performed, as shown in Table 6, there were significant correlations between antioxidant ability and beneficial lipophilic antioxidant compounds content in EDS oils [32]. There were significant correlations between lipophilic antioxidant compounds and antioxidant capacity of the EDS oils assessed by DPPH and hydroxyl radical assays (*r.* 0.728–0.915, *p* < 0.01). Thus, the above-mentioned compounds in EDS oils may have essential impact on its antioxidant ability. As shown in Table 6, the total tocopherols and tocotrienols content correlated to DPPH and hydroxyl radical values were higher than that of total phytosterols content. This represented that tocopherols and tocotrienols play a more important role than phytosterols in antioxidant capacity of EDS oils. Besides, the lupeol content correlated to DPPH and hydroxyl radical values was the highest (*r.* was 0.886 and 0.791, respectively) in all phytosterols. The possible reason we suspected was that there was a unsaturated side chain of lupeol positioned at C-21. The correlation value of γ-tocopherol content to DPPH and hydroxyl radical was the highest, which demonstrated that γ-tocopherol was the most important antioxidant in EDS oil.

## 3. Materials and Methods

### 3.1. Materials 

*E. mollis* Diels seeds were collected from Linfen, Shanxi Province, China in July, 2017. The seeds used in the experiments were ground into powder in a grinder mill (JP-3000C, Jiupin Instruments Co., Ltd., Yongkang, China) for around 1 min, and passed through 40, 60, 80 and 100 mesh sieves respectively. The seed moisture was determined by an infrared moisture analyzer (MA35M, Sartorius, Goettingen, Germany). The result showed that EDS moisture was 8.49 ± 0.021 wt.%. The total oil content of seed was measured by Soxhlet extraction (SOX406, Hanon Instruments Co., Ltd., Jinan, China) which was 32.85%. 

### 3.2. Extraction Process of EDS Oils

#### 3.2.1. Supercritical Carbon Dioxide Extraction

SF-CO_2_ extraction was performed with a Waters SFE 500 device (Shanghai, China) equipped with a 500 mL extraction vessel, high pressure pump and automated temperature regulator. Figure 3 displays a scheme of a lab supercritical fluid extractor. In all experiments, around 120 g samples were used. The SF-CO_2_ flow rate was 40 g/min constantly. During all the experimental extraction process, the extraction pressure, extraction temperature and extraction time were controlled by regulating the adjusting valves on the equipment panel. When the extraction time was achieved, depressurizing the SF-CO_2_, and the oil was collected in the separation tank.

OAD design and statistical analysis: for single factor experiment results, OAD was used to acquire the optimal SF-CO_2_ extraction conditions. Extraction pressure (A), temperature (B), time (C) and particle size (D) as four factors for three levels. Using an L_9_ (3^4^) orthogonal experimental design, investigated the influence of these factors on the EDS oil extraction yield obtained by SF-CO_2_ extraction.

#### 3.2.2. Soxhlet Extraction

Around 30 g EDS powder was put in a Soxhlet extractor and extracted with anhydrous ether at 65 °C for 4 h. Then the ether was evaporated from the extraction mixture at 80 °C for 40 min. Finally, the remaining EDS oil was collected in the extractor. 

#### 3.2.3. Cold-Press Extraction

2000 g EDS powder were cold-pressed at room temperature with the press (Model CZR091, Wuxi Dehe Industry, Wuxi, Jiangsu, China) at 60 MPa, below 30 °C for 1 h according to Willems et al. [33].

#### 3.2.4. EDS Oil Extraction Yield

The extraction yield was represented as percentage and determined by the following equation:EDS oil yield (%) = (weight of oil/weight of EDS sample powder) × 100%(1)

### 3.3. Fatty Acid Composition of EDS Oils

All the EDS oil samples were converted into fatty acid methyl esters (FAMEs) according to the method used by Bimakr [34]. Then the analysis of fatty acid composition was performed by gas chromatography (GC-2010PLUS, Shimadzu, Kyoto, Japan), equipped with a TR-FAME (30 m × 0.25 mm × 0.25 μm) capillary column and flame ionization detector (FID). Nitrogen was used as a carrier gas with a constant flow rate of 1 mL/min. The oven temperature was set as 60 °C for 3 min initially, then was raised at 5 °C/min to 175 °C and held at that for 15 min. Finally, the oven temperature increased to 220 °C at the rate of 2 °C/min and held for 10 min. Fatty acid composition was identified by comparing the retention time in samples to the FAMEs standard mixture which was purchased from Sigma-Aldrich (Shanghai, China). Each and total fatty acid were quantified as percentages which were calculated by the normalization method.

### 3.4. Lipophilic Antioxidant Compounds in EDS Oils

Total tocopherols and tocotrienols content were measured by a Waters HPLC system equipped with a Waters 2487 ultraviolet detector set at an absorbance of 290 nm and a Luna Silica column (150 mm × 4.6 mm, 3 μm particle size) used at 35 °C. The test method was improved based on the method Boso et al. [35] used. The mobile phase consisted of 0.2% IPA (Isopropanol) and 0.8% acetic acid in *n*-hexane flowing at 1.0 mL/min. Extracted oil samples (1 g) were diluted with 10 mL *n*-hexane and injected into the HPLC system.

Preparation of unsaponifiable fraction was used to determine the phytosterols composition following the method used by Iafflice et al. [9,36]. GC/MS analysis was performed using an Agilent DB-5 capillary column (30 m × 0.25 mm × 0.25 μm). Helium was used as the carrier gas with a flow rate of 1 mL/min. The initial oven temperature was held at 200 °C for 0.5 min then raised to 300 °C at 10 °C/min for 20 min. The transfer line temperature and ion source temperature were 280 °C and 250 °C, respectively. A sample injection of 1 μL was performed in a split mode of 100:1 at 250 °C. Detection was performed in the full scan mode from *m*/*z* 50 to 650. Compounds were identified by matching mass spectra and retention times with those pure compounds. NIST Mass Spectra Library was used as a reference.

### 3.5. Antioxidant Ability Analysis

The antioxidant ability of EDS oils extracted by different extraction methods were analyzed by DPPH and hydroxyl radical scavenging assays. This assay was improved and accomplished as reported by previous research. The inhibitory effect of EDS oils was expressed as percentage which was used as an index to assessed the antioxidant activities.

The DPPH radical scavenging ability of EDS oil was tested according to the method described by Xu et al. [37] 2 mL, 0.2 mM methanolic DPPH solution was mixed with EDS oil that diluted in isopropanol. The reaction was kept for 30 min in the dark and measured the absorbance at 515 nm on a spectrophotometer. The inhibition percentage of DPPH was calculated as the following formula:% DPPH radical inhibition ratio = (A_C_ − A_S_)/A_C_ × 100%(2)
where A_C_ represents the absorbance value of control sample, and A_S_ represented the absorbance value of oil sample.

Hydroxyl radical scavenging capacity of EDS oils was determined following the method performed by Kaur [38]. The reaction system (10 mL) contained 1 mL (9.0 mM) FeSO_4_·7H_2_O solution, 1 mL (8.8 mM) H_2_O_2_ solution and 1 mL (9.0 mM) salicylic acid ethanol solution with 7 mL oil sample (1, 2, 3, 4, 6, 8 and 10 mg/mL). After incubation at 37 for 30 min, the absorbance was measured at 510 nm on a spectrophotometer. The inhibition percentage of hydroxyl radical assay was calculated as the following formula:% hydroxyl radical inhibition ratio = (A_C_ − A_S_)/A_C_ × 100%(3)
where A_C_ represents the absorbance value of control sample, and A_S_ represents the absorbance value of the oil sample.

## 4. Conclusions

The optimization of EDS oil yield SF-CO_2_ extraction parameters used OAD method. The maximum EDS oil yield was achieved at 30 MPa, 50 °C, 150 min and 80 mesh. Compared with SE and CP extraction methods, SF-CO_2_-extracted EDS oil presented better physicochemical characteristics (higher linoleic acid, oleic acid, palmitic acid, γ-tocopherol, γ-tocotrienol and total phytosterols content). Furthermore, the antioxidant ability of EDS oils obtained by different extraction methods were tested by DPPH and hydroxyl radical scavenging assays. There were no significant differences on the antioxidant ability of EDS oil extracted by different methods. Above all the results, SF-CO_2_-extracted EDS oil has properties of antioxidant and health-caring with development potential for human health and functional food industry.

## Figures and Tables

**Figure 1 molecules-24-00911-f001:**
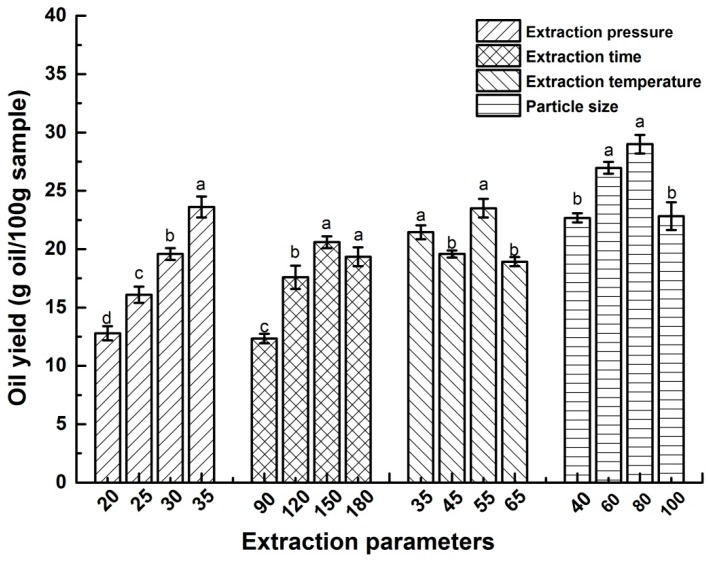
EDS oil yield at different extraction parameters. Values in the bar with different letters (a, b, c, d) are significant difference at *p* < 0.05.

**Figure 2 molecules-24-00911-f002:**
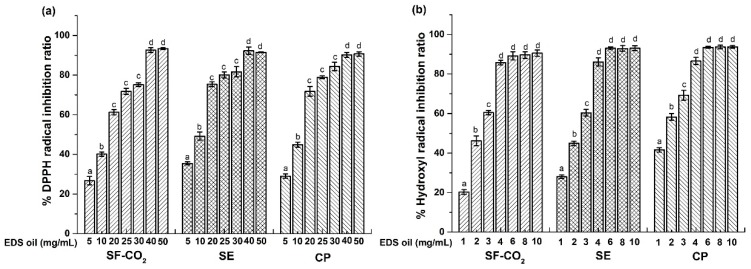
Free radical scavenging activities of EDS oil determined by (**a**) DPPH and (**b**) Hydroxyl assays. Data represent the mean of three independent experiment, each performed in triplicate (n = 3). Values in the bar with different letters (a, b, c, d) are significant difference at *p* < 0.05.

**Figure 3 molecules-24-00911-f003:**
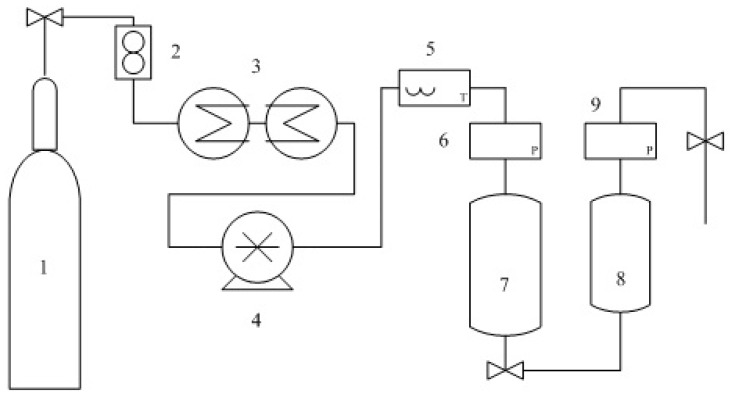
Scheme of the experimental set-up for supercritical CO_2_ fluid extraction. 1: CO_2_ feed tank; 2: flow meter; 3: cryostat; 4: pump; 5: heater; 6: manometer; 7: extraction tank; 8. Separation tank; 9. manometer.

**Table 1 molecules-24-00911-t001:** Assignment of the factors and levels using an OA_9_(3^4^) matrix along with results obtained for SFE of EDS oil.

Run no.	A (MPa)	B (°C)	C (min)	D (mesh)	Extraction Yield (%)
1	35 (3)	50 (1)	150 (2)	80 (2)	27.35
2	30 (1)	60 (3)	180 (3)	80 (2)	27.23
3	32.5 (2)	55 (2)	120 (1)	80 (2)	24.80
4	35 (3)	55 (2)	180 (3)	60 (1)	25.70
5	30 (1)	55 (2)	150 (2)	100 (3)	27.85
6	32.5 (2)	50 (1)	180 (3)	100 (3)	28.02
7	30 (1)	50 (1)	120 (1)	60 (1)	24.47
8	32.5 (2)	60 (3)	150 (2)	60 (1)	26.71
9	35 (3)	60 (3)	120 (1)	100 (3)	24.29
K_1_	79.55	79.84	73.56	76.88	
K_2_	79.53	78.35	81.91	79.38	
K_3_	77.34	78.23	80.95	80.16	
R	0.737	0.537	2.783	1.093	

**Table 2 molecules-24-00911-t002:** Analysis of variance (ANOVA) of the EDS oil extraction.

Factor	SS	f	MS	F	*p*
Extraction pressure	1.076	2	0.538	0.200	0.827
Extraction temperature	0.536	2	0.268	0.090	0.911
Extraction time	13.917	2	0.959	11.700	0.009
Particle size	1.957	2	0.979	0.380	0.700
Residual Error	0.008	2	0.004	1.350	
Total	17.494	8			

**Table 3 molecules-24-00911-t003:** Fatty acid composition of EDS oil extracted by different methods.

Fatty Acid	SF-CO_2_-Extracted	Soxhlet-Extracted	Cold-Pressed
Myristic (C14:0)	0.03	0.03	-
Pentadecanoic (C15:0)	0.02	-	-
Palmitic (C16:0)	4.12	3.59	7.24
Margaric (C17:0)	0.05	0.04	-
Stearic (C18:0)	2.50	2.54	1.56
Arachidic (C20:0)	0.19	0.23	0.15
Behenic (C22:0)	0.04	0.09	-
Tricosanoic (C23:0)	-	0.02	-
Tetracosanoic (C24:0)	0.06	0.16	-
Palmitoleic (C16:1 n-7)	0.07	0.07	-
Oleic (C18:1 n-9)	35.18	32.96	38.47
Eicosenoic (C20:1 n-11)	0.50	0.53	0.02
Linoleic (C18:2 n-9, 12)	50.99	46.71	51.83
Linolenic (C18:3 n-9, 12, 15)	5.10	4.84	0.57
Eicosadienoic (C20:2 n-11, 14)	0.05	0.05	0.03
SFA	7.01	6.70	8.95
MUFA	35.75	33.56	38.49
PUFA	56.14	51.60	52.43

SFA: saturated fatty acid; MUFA: monounsaturated fatty acid; PUFA: polyunsaturated fatty acid.

**Table 4 molecules-24-00911-t004:** Composition of tocopherols and tocotrienols in EDS oils (mg/100 g).

Types	SF-CO_2_ Extracted	Soxhlet-Extracted	Cold-Pressed
α-tocopherol	17.79 ± 0.11	16.51 ± 0.78	15.18 ± 1.32
β-tocopherol	-	-	-
γ-tocopherol	53.68 ± 2.21	49.00 ± 1.12	50.37 ± 0.56
δ-tocopherol	8.72 ± 0.032	8.83 ± 1.65	7.27 ± 0.12
α-tocotrienol	5.78 ± 0.013	5.34 ± 0.097	4.91 ± 0.038
β-tocotrienol	-	-	-
γ-tocotrienol	6.03 ± 0.56	3.59 ± 0.034	3.8 ± 0.041
δ-tocotrienol	2.25 ± 0.072	5.14 ± 0.16	4.59 ± 0.21
Total	96.24 ± 3.01	88.41 ± 4.32	86.12 ± 2.05

**Table 5 molecules-24-00911-t005:** Phytosterols in EDS oil extracted by different methods (mg/100 g).

Types	SF-CO_2_ Extracted	Soxhlet-Extracted	Cold-Pressed
β-Sitosterol	223.52 ± 7.15	212.40 ± 6.58	127.52 ± 2.32
Stigmasterol	54.10 ± 8.76	42.03 ± 3.56	35.81 ± 4.65
β-Amyrin	31.06 ± 0.30	34.36 ± 2.13	19.92 ± 6.54
Lupeol	58.66 ± 4.67	74.19 ± 5.87	50.85 ± 3.19
Total	364.34 ± 4.86	362.98 ± 2.03	224.13 ± 1.93

**Table 6 molecules-24-00911-t006:** Correlation between free radical scavenging capacity and beneficial lipophilic antioxidant compounds content in EDS oils.

Lipophilic Antioxidant Compounds	DPPH	Hydroxyl
Total phytosterols	0.860 **	0.765 **
β-Sitosterol	0.829 **	0.737 **
Stigmasterol	0.854 **	0.768 **
β-Amyrin	0.844 **	0.750 **
Lupeol	0.886 **	0.791 **
Total tocopherols and tocotrienols	0.913 **	0.824 **
α-Tocopherol	0.909 **	0.819 **
γ-Tocopherol	0.915 **	0.829 **
δ-Tocopherol	0.907 **	0.813 **
α-Tocotrienol	0.909 **	0.818 **
γ-Tocotrienol	0.798 **	0.728 **
δ-Tocotrienol	0.815 **	0.750 **

** means *p* < 0.01.

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
