# Peer review of "Supercritical CO2 Fluid Extraction of Elaeagnus mollis Diels Seed Oil and Its Antioxidant Ability"

_molecules, 2019, doi:10.3390/molecules24050911_

Round 1
Reviewer 1 Report
The manuscript by Wang et al. (article ID molecules-443452) describes the optimization of lipids from Elaeagnus mollis Diels seed by supercritical CO2 and evaluated the antioxidant ability. The research methdology and data interpretation was very strong. The quality of the manuscript is further improved by correcting the following points.
- Can you determine the position of the fatty acid double bond? (Eg. C16:1 => C16:1n-9 or C16:1n-7)
- 3.1 Materials, You should describe the information of the sample in more detail, e.g., collectiuon season, the method for ground, and total oil content.
- L252, Please describe the manufacturer of the standard mixtures you used. On the other hand, why did you use GC/MS but GC/FID?
- L 230, Why did you use ether in Soxhlet extraction? Since the polarity is near, hexane is often used as the extraction solvent when compared with supercritical extraction.
Author Response
Responses to reviewer 1:
Comments 1: Can you determine the position of the fatty acid double bond? (Eg. C16:1 => C16:1n-9 or C16:1n-7)
Reply: Thank you very much for your suggestion. According to the instruction of standard mixtures, we have carefully revised the manuscript in section 2.3 of revised manuscript and shown as following: C16:1 n-7, C18:1 n-9, C20:1 n-11, C18:2 n-9, 12, C18:3 n-9, 12, 15, C20:2 n-11,14.
Comments 2: 3.1 Materials, You should describe the information of the sample in more detail, e.g., collection season, the method for ground, and total oil content.
Reply: Thanks very much for your good suggestion. According to the suggestion of reviewer, the materials information was added in the lines 232-234, 236-237.
Comments 3: L252, Please describe the manufacturer of the standard mixtures you used. On the other hand, why did you use GC/MS but GC/FID?
Reply: Thank you very much for your kind suggestion. We have added the manufacturer of major instrument. The fatty acid methyl esters (FAMEs) standard mixture information was added in the lines 274-275 of revised manuscript and shown as following:
Fatty acid composition was identified by comparing the retention time in samples to the FAMEs standard mixture which was purchased from Sigma-Aldrich (Shanghai, China).
Compared with GC/MS, GC/FID has been proved to be a quick, simple and accurate detection method used in the analysis of fatty acid composition. Moreover, we used GC/FID to determine the composition of fatty acid in samples which was described in the lines 269-270. The preparation method of FAMEs and GC/FID detection method used in our experiment were similar with many other literatures (Subroto., et al., Journal of food science and technology, 2017; Shao., et al., European Journal of Lipid Science and Technology, 2015; Long., et al., Bioresource Technology, 2011). Fatty acid composition was identified by comparing the retention time in samples to the FAMEs standard mixture clearly.
Comments 4: L 230, Why did you use ether in Soxhlet extraction? Since the polarity is near, hexane is often used as the extraction solvent when compared with supercritical extraction.
Reply: Yes, hexane was widely used in Soxhlet extraction. The aim of Soxhlet extraction was to evaluate the bioactive chemical composition in E. mollis Diels seed oil and compared with other extraction methods. Although the polarity of ether is higher than hexane and SF-CO2, ether was also considered as the common solvent in oil extraction completely. Based on our preliminary experiment result, we found that the unsaponifiable matter content extracted by ether and hexane was around 0.8% and 1.1%, respectively. Finally, we used ether as solvent in Soxhlet extraction. What’s more, previous literature has similar conclusion (Liu, X., et al., Journal of Essential Oil Bearing Plants, 2017).
Reviewer 2 Report
The authors of manuscript entitled "Supercritical CO2 fluid extraction of Elaeagnus mollis Diels seed: Optimization and Its antioxidant ability" deal with extraction protocol of bioorganic matrix (mainly fatty acids) from Elaeagnus mollis Diels seed and measurement of antioxidants activity of EDS oil using common DPPH sensor. In my opinion this manuscript seems more as simple analytical protocol paper with very minor scientific novelty. I cannot positively rank this manuscript for publication in Molecules due to the fact that this work may be classified as the MPU paper (accordingly to Anal. Bioanal. Chem., 2016, 408:663-664.), also considering the large number of funding sources covering the research of this simple experimental work.
Author Response
Responses to reviewer 2:
Comments 1: In my opinion this manuscript seems more as simple analytical protocol paper with very minor scientific novelty. I cannot positively rank this manuscript for publication in Molecules due to the fact that this work may be classified as the MPU paper (accordingly to Anal. Bioanal. Chem., 2016, 408:663-664.), also considering the large number of funding sources covering the research of this simple experimental work.
Reply: Thank you very much for your comments. According to your suggestions, we have added experiments in the effects of supercritical CO2 fluid extraction parameters on the oil yield to strengthen the optimization of supercritical CO2 fluid extraction conditions. There were four highlights in our revised manuscript as following:
1. E. mollis Diels is regard as a rare species in China. There were few studies focused on the extraction of E. mollis Diels seed oil. For the health benefits of E. mollis Diels seed oil, it has great potential in pharmaceutical and functional food applications. Our results would provide guidelines to the further commercial development of E. mollis Diels seed oil.
2. Especially, it was the first time to find lupeol in E. mollis Diels seed oil. Compared with the literature that lupeol content in E. mollis Diels seed oil was about 5-6 times higher than that in sea buckthorn seeds oil (Li T S C. et al., Food Chemistry, 2007).
3. The correlation between the antioxidant ability of E. mollis Diels seed oil and phytosterols, tocopherols and tocotrienols content was evaluated which was proved that γ-tocopherol was the most important antioxidant in E. mollis Diels seed oil.
4. Based on our results, the E. mollis Diels seed oil extracted by supercritical CO2 fluid contained various antioxidants so that it may have properties of antioxidant and health-caring with development potential for human health and functional food industry.
Furthermore, the revised manuscript could provide the details facets of the supercritical CO2 fluid extraction of E. mollis Diels seed oil. Besides, the results could provide some important information for the further commercial development of E. mollis Diels seed oil.
Reviewer 3 Report
The manuscript by Wang et al. describes the optimization of lipids from Elaeagnus mollis Diels seed by supercritical CO2 and evaluated the antioxidant ability. Few points need to be improved to better qualify the article:
Details on the material (season, oil content, etc)
Provide any evidence on the use of hexane for the extraction of lipids. This is the most common solvent and is avoided of dangerous peroxides in ether that could modify unsaturated FA.
Author Response
Responses to reviewer 3:
Comments 1: Details on the material (season, oil content, etc)
Reply: Thanks very much for your kind suggestion. According to the comment of reviewer, we have added the collection season, the ground method and the total oil content of materials which were shown in the lines 232-234, 236-237.
Comments 2: Provide any evidence on the use of hexane for the extraction of lipids. This is the most common solvent and is avoided of dangerous peroxides in ether that could modify unsaturated FA.
Reply: Thank you very much for your comment. Yes, hexane was widely used in Soxhlet extraction. The aim of Soxhlet extraction was to evaluate the bioactive chemical composition in E. mollis Diels seed oil and compared with other extraction methods. Although the polarity of ether is higher than hexane and supercritical CO2 fluid, ether was also considered as the solvent in oil extraction completely. Based on our preliminary experiment result, we found that the unsaponifiable matter content extracted by ether was higher than that extracted by hexane. Finally, we used ether as solvent in Soxhlet extraction. What’s more, previous literature has similar conclusion (Liu, X., et al., Journal of Essential Oil Bearing Plants 2017).
Round 2
Reviewer 2 Report
The authors added some additional information from literature to the improved manuscript (R1). In my opinion there is still the problem with the novelty and general concept of MPU papers.